# The Light- and Jasmonic Acid-Induced *AaMYB108-like* Positive Regulates the Initiation of Glandular Secretory Trichome in *Artemisia annua* L.

**DOI:** 10.3390/ijms241612929

**Published:** 2023-08-18

**Authors:** Hang Liu, Weizhi He, Xinghao Yao, Xin Yan, Xiuyun Wang, Bowen Peng, Yaojie Zhang, Jin Shao, Xinyi Hu, Qing Miao, Ling Li, Kexuan Tang

**Affiliations:** Frontiers Science Center for Transformative Molecules, Joint International Research Laboratory of Metabolic and Developmental Sciences, Plant Biotechnology Research Center, Fudan-SJTU Nottingham Plant Biotechnology R&D Center, School of Agriculture and Biology, Shanghai Jiao Tong University, Shanghai 200240, China; liu-hang@sjtu.edu.cn (H.L.);

**Keywords:** *AaMYB108-like*, glandular trichome initiation, light, jasmonic acid, *Artemisia annua*

## Abstract

The plant *Artemisia annua* L. is famous for producing “artemisinin”, which is an essential component in the treatment of malaria. The glandular secretory trichomes (GSTs) on the leaves of *A. annua* secrete and store artemisinin. Previous research has demonstrated that raising GST density can effectively raise artemisinin content. However, the molecular mechanism of GST initiation is not fully understood yet. In this study, we identified an MYB transcription factor, the *AaMYB108-like*, which is co-induced by light and jasmonic acid, and positively regulates glandular secretory trichome initiation in *A. annua*. Overexpression of the *AaMYB108-like* gene in *A. annua* increased GST density and enhanced the artemisinin content, whereas anti-sense of the *AaMYB108-like* gene resulted in the reduction in GST density and artemisinin content. Further experiments demonstrated that the *AaMYB108-like* gene could form a complex with AaHD8 to promote the expression of downstream *AaHD1*, resulting in the initiation of GST. Taken together, the *AaMYB108-like* gene is a positive regulator induced by light and jasmonic acid for GST initiation in *A. annua.*

## 1. Introduction

Malaria is a global health problem with a high-risk population of more than 3 billion people, mainly in Southeast Asia and Africa [1,2]. Artemisinin combination therapy (ACT) is the only effective malaria treatment recognized by the World Health Organization [3]. Artemisinin is a sesquiterpenoid isolated from the traditional Chinese medicine *Artemisia annua*, which is mainly synthesized and stored in the peltate glandular secretory trichome on the surface of *A. annua* leaves [4,5]. In wild-type *A. annua*, the artemisinin content is only 0.1–1.0% of the dry weight; therefore, research on the biosynthesis pathway of artemisinin and the development of glandular secretory trichome is important to improve the content of artemisinin in *A. annua* [6].

As a specialized metabolite, the biosynthetic pathway of artemisinin has been fully resolved [6]. Studies surrounding the transcriptional regulation of four key artemisinin enzymes, ADS, CYP71AV1, DBR2, and ALDH1, have become a hot topic in recent years. Researchers have achieved many results in the study of transcription factors promoting artemisinin biosynthesis. AaERF1, AaERF2, and AaTAR1, all belonging to the AP2/ERF transcription factor family, enhance artemisinin biosynthesis by promoting the expression of *ADS*, *CYP71AV1*, and *DBR2* [7,8,9]. Artemisinin biosynthesis is regulated by several phytohormones. Jasmonic acid and abscisic acid have been found to promote artemisinin biosynthesis. bZIP transcription factor AabZIP1 is regulated by ABA to promote artemisinin biosynthesis [10]. AaORA and AaGSW1 (GLANDULAR TRICHOME-SPECIFIC WRKY 1) are transcription factors that are induced by both jasmonic acid and abscisic acid. AaORA is a positive regulator of artemisinin synthesis, and AaGSW1 not only promotes *CYP71AV1* expression directly but also promotes artemisinin biosynthesis indirectly through AaORA [8,11]. AaTCP15 can form a complex with AaORA and thus promote the expression of downstream genes [12]. Light signals, as the most important signals for plants in nature, have profound effects on plant growth, development, and specialized metabolism [13,14,15,16]. It was found that light signaling can co-regulate artemisinin synthesis with jasmonic acid signaling. The WRKY family transcription factor, AaWRKY9, is co-induced by light and jasmonic acid to promote artemisinin biosynthesis [17]. AaMYB108 reveals a light-dependent molecular mechanism of jasmonic acid-promoted artemisinin biosynthesis through interactions with AaGSW1 and AaCOP1 at the protein level [18].

The glandular secretory trichome on the surface of A. annua leaves is the main site of artemisinin biosynthesis and storage; therefore, increasing the density of the glandular secretory trichome is important to increase artemisinin content [19]. The HD-ZIP family transcription factors, AaHD1 and AaHD8, are positive regulators of *A. annua* glandular secretory trichome initiation. AaHD8 directly binds to the promoter of *AaHD1* to promote the expression of *AaHD1*, thereby promoting the initiation of glandular secretory trichomes [20,21]. The HD-ZIP family is considered to be the main module of glandular secretory trichome development in *A. annua*. Other families of transcription factors have also been found to promote the initiation of *A. annua* glandular secretory trichomes. The MYB transcription factor AaMYB16 forms a complex with AaHD1 to promote the expression of downstream genes [22]. ERF family transcription factor, AaWIN1, is a positive regulator of glandular secretory trichome initiation [23]. AaSPL9 can directly regulate the expression of HD1 and thus promote the initiation of glandular secretory trichomes [24].

MYB family transcription factors play important roles in plant development and metabolism. Transcriptome analysis of co-treatment with light and jasmonic acids revealed that MYB family transcription factors were involved in the process of regulation of artemisinin biosynthesis by light and jasmonic acids in *A. annua* seedlings [4]. AaMYB108 is the core factor integrating light and jasmonic acid signaling to regulate artemisinin biosynthesis in *A. annua* [18]. There is evidence that homologs of AaMYB108 are present in *A. annua*, and it is interesting to study the function of AaMYB108 homologs.

According to previous studies, glandular trichome initiation is also regulated by hormones and light, but the molecular mechanism by which light and hormones co-regulate glandular secretory trichome development in *A. annua* is unclear. In this study, we identified an MYB transcription factor, the *AaMYB108-like* gene, as a positive regulator of the initiation of glandular secretory trichomes. The *AaMYB108-like* gene was co-induced by light and jasmonic acid. Overexpression of the *AaMYB108-like* gene increased glandular secretory trichome density, and thus the *AaMYB108-like* gene is a positive regulator of glandular secretory trichome development in *A. annua*. Further experiments demonstrated that the *AaMYB108-like* gene could form a complex with AaHD8 to promote the expression of downstream *AaHD1* and, thus, the initiation of GST. Taken together, we identified an MYB transcription factor that is co-induced by light and jasmonic acid, which could promote artemisinin accumulation by positively regulating the GST density in *A. annua* leaves.

## 2. Results

### 2.1. Identification of the AaMYB108-like Gene from A. annua

AaMYB108 is a positive regulator of artemisinin biosynthesis that is co-induced by light and jasmonic acid. We found the homolog gene of AaMYB108 by sequence analysis, Aannua01621S233630, which is named the *AaMYB108-like* gene (Figure 1A). We further investigated whether the *AaMYB108-like* gene is involved in artemisinin synthesis and glandular secretory trichome development. By gene cloning, we obtained a gene with a coding sequence (CDS) length of 960 bp, encoding 319 amino acids. The amino acid sequence comparison revealed that the *AaMYB108-like* gene was 60% homologous to AaMYB108 (Figure 1B).

### 2.2. Co-Induction of AaMYB108-like Gene Expression by Light and Jasmonic Acid

To further characterize the *AaMYB108-like* gene, we examined the effects of light and jasmonic acid on its expression. We used 100 µM of exogenous methyl jasmonate (MJ) for wild-type *A. annua* seedlings and analysis of the quantitative real-time PCR (qRT-PCR) results revealed that *AaMYB108-like* gene expression was induced by jasmonic acid (Figure 2A). Similarly, we subjected wild-type *A. annua* seedlings to light and dark treatments, and the qRT-PCR results showed that light could induce the expression of the *AaMYB108-like* gene (Figure 2B). According to the above experimental results, the *AaMYB108-like* gene is an MYB family transcription factor induced by both light and jasmonic acid.

### 2.3. Expression Pattern of the AaMYB108-like Gene

According to the transcriptome data of the various tissues in *A. annua*, glandular trichomes are where the *AaMYB108-like* gene is highly expressed. QRT-PCR research was carried out to look into the *AaMYB108-like* gene expression patterns in both space and time. The *AaMYB108-like* gene had the highest expression in glandular trichomes in various tissues, according to the results of the qRT-PCR study (Figure 3A).

In order to more clearly illustrate the expression pattern of the *AaMYB108-like* gene, we transformed wild-type (WT) *A. annua* with 1391Z-proMYB108-like-GUS (-glucuronidase), where expression of the GUS reporter is driven by the 1878-base pair (bp) promoter sequence of the *AaMYB108-like* gene. Histochemical GUS staining of the 1391Z-proMYB108-like-GUS transgenic lines revealed GUS activity, particularly in the glandular secretory trichomes (GST) of early leaves and stems (Figure 3B). The expression pattern of the GUS reporter and the detected expression patterns by qRT-PCR exhibited a good correlation. In summary, the *AaMYB108-like* gene is a transcription factor that is highly expressed in *A. annua* glandular trichomes.

### 2.4. Subcellular Localization of the AaMYB108-like Gene

We created a pHB-AaMYB108-like-YFP (yellow fluorescent protein) fusion construct and transiently expressed it in *Nicotiana benthamiana* leaves in order to further investigate the subcellular location of the *AaMYB108-like* gene (Figure 4). We discovered via fluorescence microscopy that the YFP signals were very strong in the nucleus. According to these findings, the AaMYB108-like protein is localized to the nucleus, which is consistent with its function as a transcription factor.

### 2.5. Overexpressing AaMYB108-like Gene Increased GST Density in A. annua

Transgenic *A. annua* plants were created in order to investigate the function of the *AaMYB108-like* gene. After PCR detection, we obtained 17 *AaMYB108-like* overexpression lines and 14 anti-*AaMYB108-like* lines. Following the transgenic test, three *AaMYB108-like* overexpression lines and three anti-*AaMYB108-like* lines were chosen for phenotype analysis. Firstly, the leaves of *AaMYB108-like* transgenic plants were collected for artemisinin extraction, and the content of artemisinin in the leaves was determined by HPLC. The results of the HPLC assay showed that overexpression of the *AaMYB108-like* gene increased the artemisinin content compared to wild-type *A. annua*, while the interference expression of the *AaMYB108-like* gene decreased the artemisinin content (Figure 5A). When the expression of key enzymes in *AaMYB108-like* transgenic plants was examined, it was found that the expression of the four key enzymes of artemisinin biosynthesis did not show a trend consistent with the artemisinin content (Figure 5B,C).

Considering that the increase in the number of GST of *A. annua* can cause an increase in artemisinin content, the GST on the leaf surface of *AaMYB108-like* transgenic plants were observed in this study. Surprisingly, in lines overexpressing the *AaMYB108-like* gene, the density of GST on the surface of leaves rose by at least 60% (Figure 5D,E).

### 2.6. The AaMYB108-like Gene Interacts with AaHD8 to Promote AaHD1 Expression

To investigate the molecular mechanism by which the *AaMYB108-like* gene promotes glandular secretory trichome initiation, we used a yeast two-hybrid assay first. The results show that the *AaMYB108-like* gene interacts with AaHD8 (Figure 6A), a reported positive regulator of glandular secretory trichome initiation, which promotes downstream *AaHD1* expression. Subsequently, we used a Dual-LUC assay to confirm that the interaction of the *AaMYB108-like* gene with AaHD8 could enhance the activation of AaHD8 for AaHD1 (Figure 6B). We also examined the expression of *AaHD1* in *AaMYB108-like* transgenic plants and found that overexpression of the *AaMYB108-like* gene increased the expression of *AaHD1* (Figure 6C). Therefore, we can conclude that the *AaMYB108-like* gene forms a complex with AaHD8 to promote the expression of the downstream gene *AaHD1* and thus promote glandular secretory trichome initiation.

## 3. Discussion

Based on these results, the AaMYB108-like gene is a positive regulator that promotes glandular trichomes initiation in *A. annua*. The functions of transcription factors are usually diverse. According to the analysis of the *AaMYB108-like* gene expression pattern, we can speculate that the *AaMYB108-like* gene may be associated with flower development. Overexpression of the *AaMYB108-like* gene can increase the density of glandular trichomes, ultimately leading to the overaccumulation of artemisinin. This suggests that increasing the density of glandular trichomes is an effective strategy to increase the content of artemisinin. Light and jasmonic acid regulate the initiation of glandular trichomes via the *AaMYB108-like* gene in *A. annua*, suggesting that glandular trichomes initiation is a complex process. Interaction between the *AaMYB108-like* gene and AaHD1 explains the molecular mechanism by which the *AaMYB108-like* gene regulates glandular trichomes initiation; however, the regulatory network for glandular trichomes initiation and development remains unclear in *A. annua*.

The study of increasing the active ingredients of medicinal plants has become a desirable topic in the field of plant-specialized metabolism research. Although the exogenous application of plant growth regulators and ploidy breeding are used successfully to increase the contents of specialized metabolites in medicinal plants, genetic engineering has been demonstrated to be another effective approach [25,26,27,28,29,30,31]. For example, overexpression of the biosynthetic pathway genes or downregulation of the competitive branching pathway by anti-sense or RNAi technologies can improve the production of important metabolites such as artemisinin in *A. annua* [32,33,34,35], vinblastine in *Catharanthus roseus* [36], scopolamine in *Atropa belladonna* [37], and tropane alkaloids in *Hyoscyamus niger* [38]. Along with the large amount of omics data generated and the development of the regulatory network of important specialized metabolites from medicinal plant species [39,40], a range of transcription factors belonging to various families, including MYB [41], AP2/ERF [42], MYC [43], bZIP [44], WRKY [45], YABBY [46,47], TCP [48], NAC [49], SPL [50], MADS-box [51], and bHLH [52,53] have been identified and demonstrated to have positive functions for elevating the accumulation of various specialized metabolites such as artemisinin, showing their promising application values for the trait improvement in medicinal plants.

Plant glandular trichomes are known as plant factories because of their richness in natural chemical products. Therefore, increasing the density of glandular trichomes is an effective means to obtain high-content varieties. The molecular mechanism of glandular hair initiation and development remains to be studied, and no structural genes related to glandular hair initiation have been reported. In the case of *A. annua*, the study of glandular trichome initiation is also of great importance. After years of research, a regulatory network of glandular trichome initiation has been formed with HD-ZIP and MYB transcription factors as the core factors [45]. Summarizing past studies, we can find that the development of plant glandular trichomes is regulated by both hormones and light. JA, Gibberellic Acid (GA), ABA, and other plant hormones have been reported to regulate glandular trichome initiation [21,24]. However, the molecular mechanisms by which light and plant hormones jointly regulate glandular trichome initiation are not clear. On this basis, we found that the *AaMYB108-like* gene was regulated by light and jasmonic acid and further elucidated the molecular mechanisms by which light and jasmonic acid regulate glandular trichome initiation. Our study provides new ideas for exploring the molecular mechanisms of glandular trichome development. It is easy to see from our study that increasing the density of glandular trichomes can effectively increase the content of artemisinin. Our research provides new ideas for obtaining high-content varieties in production.

## 4. Materials and Methods

### 4.1. Plant Materials and Growth Conditions

The *A. annua* used in this investigation was “Huhao 1”, which was created in Shanghai following years of selection and was taken from Chongqing, China [6]. Plantlets of *A. annua* and *Nicotiana benthamiana* were planted in a glasshouse with a photoperiod of 16 h:8 h, light:dark, and 25 °C. All the plants were cultivated with a mixture of vermiculite and nutrient soil.

### 4.2. Treatment of Light, Dark and MJ

For later usage, seedlings of wild-type *A. annua* that were 14 days old and at the same growth stage were chosen. Half (10 pots, 4 plants per pot) were kept under continuous illumination and half (10 pots) were transferred to completely dark conditions. Leaves were sampled at 0 h and 3 h during treatment and frozen in liquid nitrogen for RNA extraction. For jasmonic acid treatment, 14-day-old *A. annua* seedlings with the same growth status were treated with methyl jasmonate (MJ) of 100 μmol/L, then sampled at 0 h, 1 h, 3 h, and 6 h after spraying treatment.

### 4.3. Isolation of Glandular Trichomes

Flower buds (20 g) were mixed with 0.5 mm glass beads (50 g) and extraction buffer (250 mL). The mixture was shaken thoroughly for 1 min; then, the process was repeated three times. The mixture was sequentially passed through 350 μm, 105 μm, and 20 μm nylon sieves to isolate the glass beads and tissue fragments. Glandular trichomes were collected on the 20 μm nylon sieves and were washed by buffer at least eight times. Tissues and glandular trichomes were stored and disposed of in ice.

### 4.4. RNA Extraction and RT-qPCR

Total RNA was extracted from each sample using the RNAprep Pure Plant Kit (Tiangen, Beijing, China) according to the manufacturer’s instructions in order to analyze the expression patterns of the *AaMYB108-like* gene in various tissues (root, stem, young leaf (YL), old leaf (OL), buds, flower, shoot, and trichome). PrimeScript™ RT Master Mix was used to create first-strand cDNA for qRT-PCR from whole RNA (TaKaRa, Shiga, Japan). TB Green^®^ Premix EX Taq™ II was used to conduct QRT-PCR analyses (Takara, Japan). For each analysis, three technical and biological replicates were duplicated. Actin was chosen as the reference or control. All primers used are listed in Appendix A.

### 4.5. Subcellular Localization

To create a pHB-AaMYB108-like-YFP fusion protein, the open reading frame (ORF) of the *AaMYB108-like* gene was amplified using KOD (*Thermococcus kodakaraensis* KOD1) plus DNA polymerase and then cloned into the plant expression vector pHB-YFP. The plasmid was then introduced into the *Agrobacterium tumefaciens* strain GV3101 for transient expression in *N. benthamiana* leaves, with pHB-YFP serving as the negative control. Confocal laser microscopy was used to study the fluorescent signals of *N. benthamiana* leaves after 48 h of low light (Leica TCS SP5-II).

### 4.6. Transformation of Artemisia annua

Using KOD plus DNA polymerase (Toyobo, Osaka, Japan), the 960 bp full-length cDNA of the *AaMYB108-like* gene was amplified before being cloned into the pHB vector. In order to genetically transform the *A. tumefaciens* strain EHA105 into *A. annua* for further investigation, the constructed pHB-AaMYB108-like gene was added [54]. *A. annua* seeds were first planted on germination medium MS0, and they were then grown at 24 °C to 26 °C with a 16 h light cycle and an 8 h dark cycle (8000 lux). After two weeks, the seedlings’ leaves were harvested, cut into 0.5-cm-diameter discs, and co-cultivated with the *A. tumefaciens* strain EHA105 for three days at 25 °C. As soon as the leaves were moved to the selective medium MS1 (MS0 + 2.5 mg/L N6-benzoyladenine + 0.3 mg/L naphthalene-1-acetic acid + 50 mg/L hygromycin + 250 mg/L carbenicillin), we chose the antibiotic-resistant plantlets and subcultured them three times before moving them to the rooting medium MS2 (12 MS0 + 250 mg/L carbenicillin). Eventually, the rooted plantlets were transplanted to soil pots in the growth chamber after 1 month.

### 4.7. GUS Expression in 1391Z-proMYB108-like-GUS Transgenic A. annua Plants

To create 1391Z-proAaMYB108-like-GUS plants, the 1878 bp promoter region of the *AaMYB108-like* gene upstream of the initiation codon was found in the genomes of *A. annua*. The promoter of the *AaMYB108-like* gene was cloned and inserted into the pCambia1391Z vector with the GUS reporter. For the transformation of *A. annua* plants, the recombinant plasmids were converted into *A. tumefaciens* strain EHA105. As previously mentioned, GUS staining was carried out [55]. Observation of the sample was conducted with an optical microscope, adjusted to a 10× eyepiece and 20× objective. GUS activity appeared blue in the field of view.

### 4.8. Artemisinin Content Measurement

*A. annua* leaves that were 4 months old were collected and dried in an oven at 50 °C. After that, leaves were powdered, and 0.1 g of that powder were extracted twice with 2 mL of methanol using ultrasonic for 30 min at a temperature of 55 °C. The supernatants were filtered through nitrocellulose (0.22 m) after centrifuging 12,000× *g* for 10 min. Using high-performance liquid chromatography (HPLC), the artemisinin’s components were examined [56]. The filtrates were analyzed by the Waters Alliance 2695 HPLC system coupled with a Waters 2420 ELSD detector (Milford, MA, USA). The HPLC conditions were set as described previously [8]. The standard of artemisinin was purchased from Sigma, and the standards of dihydroartemisinic acid were obtained from Guangzhou Honsea Sunshine Bio Science and Technology Co., Ltd. (Guangzhou, Guangdong, China). All samples were measured three times for technical replications.

### 4.9. Bimolecular Fluorescence Complementation Assays

For bimolecular fluorescence complementation (BiFC) assays, the open reading frame of the *AaMYB108-like* gene was cloned into the pxy106-nYFP vector, and the full-length coding region of *AaHD8* was inserted into the pxy104-cYFP vector. The recombinant plasmids and the empty vectors were transferred into GV3101. Subsequently, combinations of the plasmids were co-transformed into *N. benthamiana* leaves. Confocal laser microscopy (Leica TCS SP5-II) was used to observe the YFP signals.

### 4.10. Yeast Two-Hybrid (Y2H) Assays

To create the pGADT7-AaMYB108-like gene, the whole coding sequence of the *AaMYB108-like* gene was cloned and placed into the pGADT7 vector. AaHD8 were then introduced into the pGBKT7 vector. Negative controls included the usage of the empty pGADT7 and pGBKT7 vectors. The recombinant plasmid combinations were added to yeast stain AH109. In SD/-Leu/-Trp (DDO) agar medium plates, the positive clones were grown. These plates were then switched over to SD/-Leu/-Trp/-His (TDO) and SD/-Leu/-Trp/-His/-Ade (QDO) medium plates. After three days, the outcomes were observed.

### 4.11. Dual-LUC (Dual-Luciferase Reporter) Assay

The PHB-AaMYB108-like gene and pHB-AaHD8 were transformed into *A. tumefaciens* strain GV3101 to act as an effector, and the pHB empty vector was used as a negative control. The promoter of *AaHD1* was cloned into a pGREEN II 0800 vector as a reporter and transformed into *A. tumefaciens* strain GV3101 with the help of the helper plasmid pSoup 19. For the purpose of transforming 4-week-old tobacco leaves, the effectors and reporters were combined in a 9:1 volume ratio [57]. After 48 h of low light incubation, *N. benthamiana*’s invaded leaves were discovered utilizing the Dual-Luciferase Reporter Assay System (Promega, Madison, WI, USA). The relative LUC/REN ratios were utilized to reflect the activity of the promoter after the activity of LUC was normalized to the activity of REN. Four biological repeats were performed for each sample.

## 5. Conclusions

Our study identified a positive regulator promoting glandular trichomes initiation by bioinformatics analysis in *A. annua*. The expression of the *AaMYB108-like* gene was co-induced by light and jasmonic acid, and it was highly expressed in glandular trichomes. *AaMYB108-like* transgenic experiments showed that overexpression of the *AaMYB108-like* gene promoted glandular trichomes initiation in *A. annua*. The interaction between the *AaMYB108-like* gene and AaHD1 explains the molecular mechanism by which the *AaMYB108-like* gene regulates glandular trichomes initiation. Based on our results, we can conclude that the *AaMYB108-like* gene is a positive regulator of glandular trichomes initiation induced by both light and jasmonic acid. The initiation and development of plant glandular trichomes is a complex process that deserves more in-depth study.

## Figures and Tables

**Figure 1 ijms-24-12929-f001:**
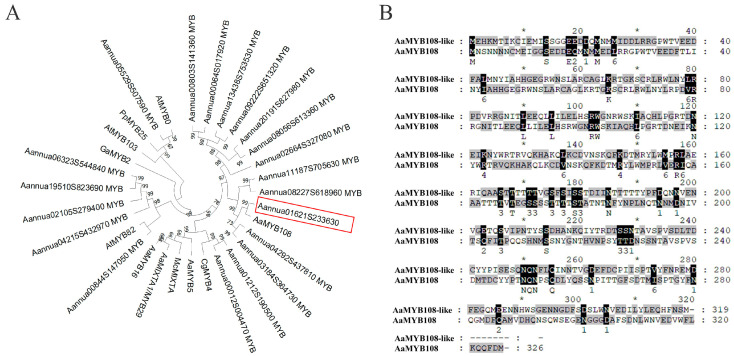
Phylogenetic analysis of *AaMYB108-like* gene. (**A**) The protein sequence alignment of *AaMYB108-like* gene and AaMYB108. Candidate genes marked by red frame. (**B**) Phylogenetic analysis was performed using MYB family proteins from *A. annua*. The tree presented here is a neighbor-joining tree based on amino acid sequence alignment and constructed using the program MEGA. Black highlights are identical amino acid sequences. Every 10 amino acids are marked with an asterisk “*”.

**Figure 2 ijms-24-12929-f002:**
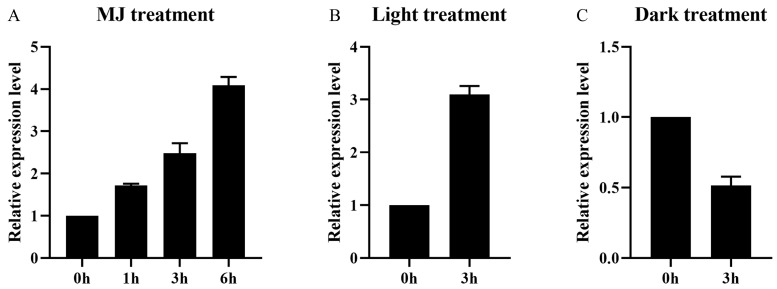
Co-induction of *AaMYB108-like* gene expression by light and jasmonic acid in *A. annua*. (**A**) Relative expression of the *AaMYB108-like* gene in response to methyl jasmonate (MJ, 100 μM) by RT-qPCR. Data values are means ± SD (*n* = 3). (**B**) Light could induce the expression of the *AaMYB108-like* gene. Data values are means ± SD (*n* = 3). (**C**) Dark treatment decreased *AaMYB108-like* expression. Data values are means ± SD (*n* = 3).

**Figure 3 ijms-24-12929-f003:**
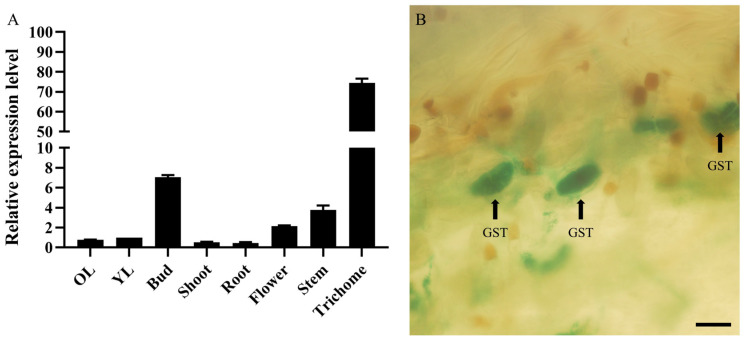
Expression pattern of the *AaMYB108-like* gene. (**A**) Relative expression levels of the *AaMYB108-like* gene in different tissues. Data values are means ± SD (*n* = 3). (**B**) β-Glucuronidase expression of 1391-proAaMYB108-like-GUS transgenic *A.annua* plants. Bars: 100 μm.

**Figure 4 ijms-24-12929-f004:**
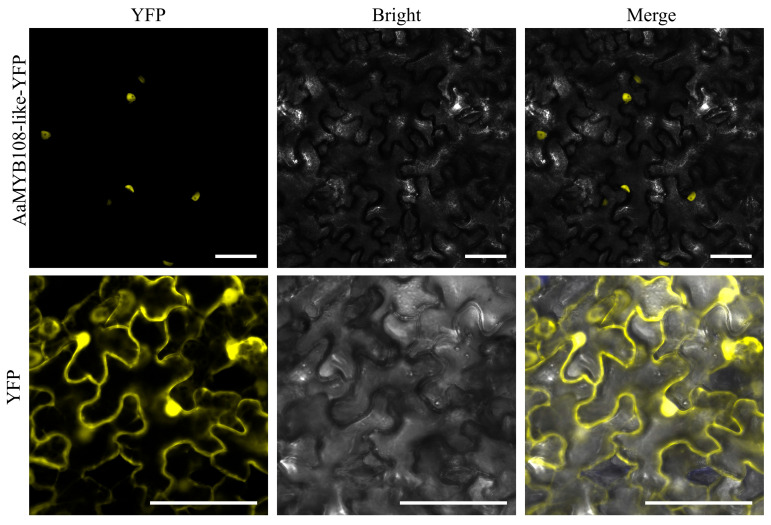
The subcellular localization of the AaMYB108-like gene in leaves of *N. benthamiana*. Yellow, yellow fluorescent protein (YFP). Bars: 50 μm.

**Figure 5 ijms-24-12929-f005:**
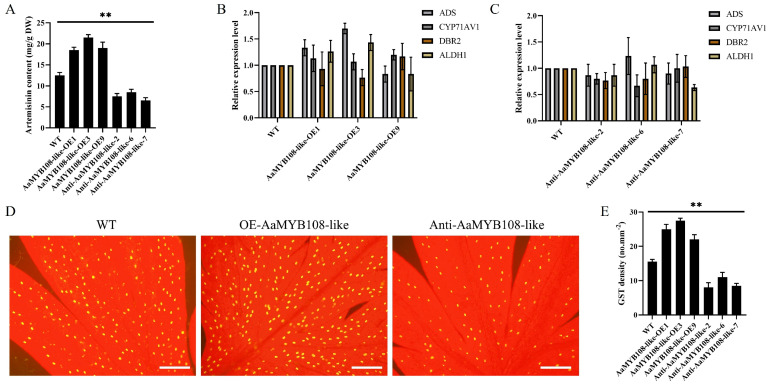
The *AaMYB108-like* gene is a positive regulator of glandular secretory trichome initiation in *A. annua*. (**A**) High-performance liquid chromatography (HPLC) analysis of the artemisinin content (mg.g^−1^ DW) in transgenic *A. annua* plants. (**B**,**C**) Expression of key enzyme genes in *AaMYB108-like* overexpression transgenic plants. (**D**) The glandular secretory trichomes on the adaxial side of mature leaves derived from wild-type (WT) plants, OE-*AaMYB108-like* transgenic *A. annua* plants, and anti-sense-*AaMYB108-like* transgenic *A. annua* plants (bars: 200 mm). (**E**) Glandular secretory trichomes density of mature leaves derived from WT and *AaMYB108-like* transgenic plants. All data are presented as means ± SD (*n* = 3) ** *p* < 0.01; Student’s *t*-test.

**Figure 6 ijms-24-12929-f006:**
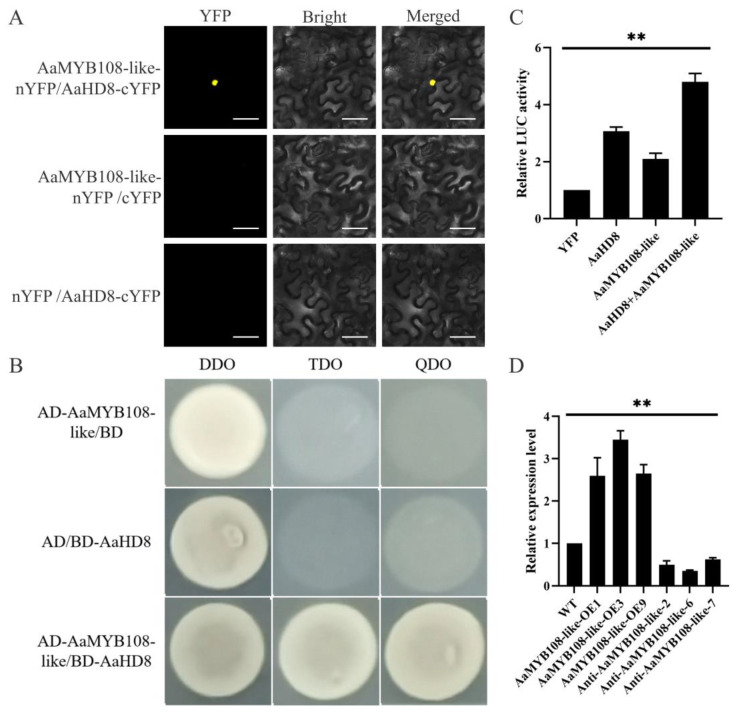
The *AaMYB108-like* gene interacts with AaHD8 to promote *AaHD1* expression. (**A**) Bimolecular fluorescence complementation assay between the *AaMYB108-like* gene and AaHD8. Yellow, yellow fluorescent protein (YFP). Bars: 50 μm. (**B**) Y2H analysis of *AaMYB108-like* gene interaction with AaHD8. (**C**) Dual-LUC assays performed in *N. benthamiana* leaves of co-expressing *AaMYB108-like* and AaHD8 genes. (**D**) Expression of *AaHD1* in AaMYB108-like transgenic plants. All data are presented as means ± SD (*n* = 3) ** *p* < 0.01; Student’s *t*-test.

## Data Availability

The datasets presented in this study can be found in online repositories. The names of the repository/repositories and accession number(s) can be found in the article/Appendix A.

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
