# Peer review of "The Light- and Jasmonic Acid-Induced AaMYB108-like Positive Regulates the Initiation of Glandular Secretory Trichome in Artemisia annua L."

_ijms, 2023, doi:10.3390/ijms241612929_

Round 1
Reviewer 1 Report
The manuscript: “Light- and jasmonic acid induced AaMYB108-like positively regulates glandular secretory trichome initiation in Artemisia annua” is dealing with the molecular mechanism of store artemisinin in glandular secretory trichomes of A. annua. This work represent the continuation of previously reported results that both light and jasmonic acid (JA) can promote the biosynthesis of artemisinin. From recently published reports, MYB transcription factor, AaMYB108, identified from transcriptome analysis of light and JA treatment is positive regulator of artemisinin biosynthesis in A. annua.
In this manuscript, authors identified that MYB transcription factor, AaMYB108-like, which is co-induced by light and JA is positively regulates glandular secretory trichome initiation in A. annua. Overexpression of AaMYB108-like in A. annua increased glandular secretory trichomes density and enhanced the artemisinin content, whereas antisense of AaMYB108-like resulted in the reduction of glandular secretory trichomes density and artemisinin content. In addition, AaMYB108-like could form a complex with AaHD8 to promote the expression of downstream AaHD1, resulting in the initiation of glandular secretory trichomes. From obtained results authors conclude that AaMYB108-like is a positive regulator induced by light and jasmonic acid also for glandular secretory trichomes initiation in A. annua.
The presented results has some merit and falls in a scope of the journal IJMS but it is not acceptable for the publication in present form.
There are three major suggestions for manuscript improvement:
1. I understand that authors are very familiar with the topic and have several published articles on the same topic in different journals and some information are so many times repeated but I think that there is require for each new manuscript include main references for statements in sentences. This is very important for reader of the article to be clear explanation of the scientific problem as well as significance of the presented results. For instance, lines 36, 37, after first sentence there have to be some references to cover statement in sentence. The same situation is in paragraph 74-80.
2. Why Disscusion section is so short?
3. The manuscript is not written according to Instructions for the IJMS journal. References must be numbered in order of appearance in the text (including citations in tables and legends) and listed individually at the end of the manuscript.
All minor typographical or spelling errors that I find I highlighted in text so authors can easily improve the manuscript text.

Author Response
Dear Editors and Reviewer,
Thank you very much for your email regarding our manuscript (ijms-2559954) entitled " Light- and jasmonic acid -induced AaMYB108-like positively regulates glandular secretory trichome initiation in Artemisia annua ", which was submitted to International Journal of Molecular Sciences. We appreciate the helpful comments and suggestions, which we have addressed as listed below.
The manuscript: “Light- and jasmonic acid induced AaMYB108-like positively regulates glandular secretory trichome initiation in Artemisia annua” is dealing with the molecular mechanism of store artemisinin in glandular secretory trichomes of A. annua. This work represents the continuation of previously reported results that both light and jasmonic acid (JA) can promote the biosynthesis of artemisinin. From recently published reports, MYB transcription factor, AaMYB108, identified from transcriptome analysis of light and JA treatment is positive regulator of artemisinin biosynthesis in A. annua.
In this manuscript, authors identified that MYB transcription factor, AaMYB108-like, which is co-induced by light and JA is positively regulates glandular secretory trichome initiation in A. annua. Overexpression of AaMYB108-like in A. annua increased glandular secretory trichomes density and enhanced the artemisinin content, whereas antisense of AaMYB108-like resulted in the reduction of glandular secretory trichomes density and artemisinin content. In addition, AaMYB108-like could form a complex with AaHD8 to promote the expression of downstream AaHD1, resulting in the initiation of glandular secretory trichomes. From obtained results authors conclude that AaMYB108-like is a positive regulator induced by light and jasmonic acid also for glandular secretory trichomes initiation in A. annua.
The presented results has some merit and falls in a scope of the journal IJMS but it is not acceptable for the publication in present form.
There are three major suggestions for manuscript improvement:
- I understand that authors are very familiar with the topic and have several published articles on the same topic in different journals and some information are so many times repeated but I think that there is require for each new manuscript include main references for statements in sentences. This is very important for reader of the article to be clear explanation of the scientific problem as well as significance of the presented results. For instance, lines 36, 37, after first sentence there have to be some references to cover statement in sentence. The same situation is in paragraph 74-80.
Response to comments: Thank you for your comments. We have checked the full text and made changes to the references. All changes are highlighted in yellow.
- Why Discussion section is so short?
Response to comments: Thank you for your comments. We have added the discussion and conclusion section. All changes are highlighted in yellow.
- The manuscript is not written according to Instructions for the IJMS journal. References must be numbered in order of appearance in the text (including citations in tables and legends) and listed individually at the end of the manuscript.
Response to comments: Thank you for your comments. We have revised the formatting of all references in the manuscript according to the author instructions.
All minor typographical or spelling errors that I find I highlighted in text so authors can easily improve the manuscript text.
Response to comments: Thank you very much for your appreciation of our work and many useful suggestions. Based on your comments we have made improvements to the manuscript. All changes are marked in the manuscript. The manuscript has been extensively revised to improve the quality of the writing and to correct the formatting errors present in the previous version.
Reviewer 2 Report
Dear Authors! The article is interesting and has practical importance. My suggestions:
1) Add the word gene after AaMYB108-like (lines 17, 18, 165, 171, 213, 219, 225, 228, 247, 325).
2) Enclose numbers of the references in square brackets.
3) I did not understand the sentence on lines 45-49. How are AaORA and AaGSW1 induced by AaORA? I think, this sentence is long. Rephrase it. Decipher for the first time AaORA.
4) Change the first and to comma (line 52).
5) Write AaHD1 (line 65), AaMYB108-like (lines 204, 221, 281, 325), latin names (lines 87, 125, 219), AaHD1 (lines 89, 281, 283) in italic.
6) These sentences are too long (lines 62-66, 274-277).
7) Check the sentence on line 83.
8) Decipher for the first time N. bentamiana (line 97). Why did you used this plant in your experiments? Write more about the soil type/content.
9) Decipher for the first time JA (line 104). How did you treat plants with MJ? Sprayed?
10) Decipher for the first time A. tumefacience (line 125), ORF (line 123), KOD (line 124), CDS (line 195), OL, YL (under the Figure 3), Dual-LUC (line 278), GA (line 322).
11) Give references to sentence on lines 322-323.
12) Describe references according instructions for authors.
Author Response
Dear Editors and Reviewer,
Thank you very much for your email regarding our manuscript (ijms-2559954) entitled " Light- and jasmonic acid -induced AaMYB108-like positively regulates glandular secretory trichome initiation in Artemisia annua ", which was submitted to International Journal of Molecular Sciences. We appreciate the helpful comments and suggestions, which we have addressed as listed below.
Dear Authors! The article is interesting and has practical importance. My suggestions:
1) Add the word gene after AaMYB108-like (lines 17, 18, 165, 171, 213, 219, 225, 228, 247, 325).
Response to comments: Thank you for your comments. We have added the word “gene”, the text is highlighted in yellow.
2) Enclose numbers of the references in square brackets.
Response to comments: Thank you for your comments. We have revised the formatting of all references in the manuscript according to the author instructions.
3) I did not understand the sentence on lines 45-49. How are AaORA and AaGSW1 induced by AaORA? I think, this sentence is long. Rephrase it. Decipher for the first time AaORA.
Response to comments: We apologized for the ambiguity caused by the writing error. We have changed this sentence; the text is highlighted in yellow. According to the reference “AaORA, a trichome-specific AP2/ERF transcription factor of Artemisia annua, is a positive regulator in the artemisinin biosynthetic pathway and in disease resistance to Botrytis cinerea”, AaORA is the original name, we didn't find an explanation for it.
4) Change the first and to comma (line 52).
Response to comments: Thank you for your comments. We have revised it; the text is highlighted in yellow.
5) Write AaHD1 (line 65), AaMYB108-like (lines 204, 221, 281, 325), latin names (lines 87, 125, 219), AaHD1 (lines 89, 281, 283) in italic.
Response to comments: Thank you for your comments. We have checked the text and changed the gene names and latin names to italics. All changes are highlighted in yellow.
6) These sentences are too long (lines 62-66, 274-277).
Response to comments: Thank you for your comments. We've shortened the sentence. All changes are highlighted in yellow.
7) Check the sentence on line 83.
Response to comments: We've changed this sentence. All changes are highlighted in yellow.
8) Decipher for the first time N. bentamiana (line 97). Why did you used this plant in your experiments? Write more about the soil type/content.
Response to comments: Nicotiana benthamiana is a commonly used plant in experiments, and both the subcellular localization experiments and the Dual-LUC experiments in this text were performed in N. benthamiana.
9) Decipher for the first time JA (line 104). How did you treat plants with MJ? Sprayed?
Response to comments: Thank you for your comments. We've deciphered it, and the treatment of plants is spraying. All changes are highlighted in yellow.
10) Decipher for the first time A. tumefacience (line 125), ORF (line 123), KOD (line 124), CDS (line 195), OL, YL (under the Figure 3), Dual-LUC (line 278), GA (line 322).
Response to comments: Thank you for your comments. We've deciphered it. All changes are highlighted in yellow.
11) Give references to sentence on lines 322-323.
Response to comments: We have given references to this sentence.
12) Describe references according instructions for authors.
Response to comments: Thank you for your comments. We have revised the formatting of all references in the manuscript according to the author instructions. Thank you very much for your appreciation of our work and many useful suggestions. Based on your comments we have made improvements to the manuscript. All changes are marked in the manuscript. The manuscript has been extensively revised to improve the quality of the writing and to correct the formatting errors present in the previous version.
Reviewer 3 Report
"Light- and jasmonic acid-induced AaMYB108-like positively regulates glandular secretory trichome initiation in Artemisia annua"
I suggest a small correction to the title:
"Light- and jasmonic acid-induced AaMYB108-like positive regulates the initiation of glandular secretory trichomes in Artemisia annua L."
In Materials and Methods in subsection 2.3. isolation of glandular trichomes, only flower bud trichomes were isolated.
Why did you isolate only these and what is the purpose of these isolates? Also, isolates from flower buds are not mentioned anywhere in the paper.
The paper states that all the research was done on leaf glandular hairs.
The Discussion is not well written, and the data presented in Figures 1-6 are not discussed. All obtained results should be discussed
A Conclusion should be written.
The literature used is adequate and most references are from the last five years.
The References are not cited according to the instructions for authors of the International Journal of Molecular Sciences.
With quality editing that includes explanations of isolates from flower buds and a well-written discussion comparing all the results obtained, the manuscript would have the potential for publication.
Author Response
Dear Editors and Reviewer,
Thank you very much for your email regarding our manuscript (ijms-2559954) entitled " Light- and jasmonic acid -induced AaMYB108-like positively regulates glandular secretory trichome initiation in Artemisia annua ", which was submitted to International Journal of Molecular Sciences. We appreciate the helpful comments and suggestions, which we have addressed as listed below.
Comments and Suggestions for Authors
"Light- and jasmonic acid-induced AaMYB108-like positively regulates glandular secretory trichome initiation in Artemisia annua"
I suggest a small correction to the title:
"Light- and jasmonic acid-induced AaMYB108-like positive regulates the initiation of glandular secretory trichomes in Artemisia annua L."
Response to comments: Thank you for your comments. After some discussion, we've decided to take your suggestion.
In Materials and Methods in subsection 2.3. isolation of glandular trichomes, only flower bud trichomes were isolated.
Why did you isolate only these and what is the purpose of these isolates? Also, isolates from flower buds are not mentioned anywhere in the paper.
The paper states that all the research was done on leaf glandular hairs.
Response to comments: Thank you for your comments. Just like you said, we study the initiation of leaf glandular trichomes, we need to know whether the candidate gene is expressed in the glandular trichomes. So, the purpose of isolating trichomes is for the extraction of RNA, and the qRT-PCR for candidate genes. In practice, however, we found that the glandular trichomes in the buds were more easily isolated compared to those on the leaves. Actually, in the published reports, glandular trichomes isolated from flower buds have been widely used in the study of initiation of glandular trichome in Artemisia annua L.
The Discussion is not well written, and the data presented in Figures 1-6 are not discussed. All obtained results should be discussed
A Conclusion should be written.
Response to comments: Thank you for your comments. We have added the discussion and conclusion section. All changes are highlighted in yellow.
The literature used is adequate and most references are from the last five years.
The References are not cited according to the instructions for authors of the International Journal of Molecular Sciences.
Response to comments: Thank you for your comments. We have revised the formatting of all references in the manuscript according to the author instructions.
With quality editing that includes explanations of isolates from flower buds and a well-written discussion comparing all the results obtained, the manuscript would have the potential for publication.
Response to comments: Thank you very much for your appreciation of our work and many useful suggestions. Based on your comments we have made improvements to the manuscript. All changes are marked in the manuscript. The manuscript has been extensively revised to improve the quality of the writing and to correct the formatting errors present in the previous version.
Round 2
Reviewer 1 Report
Authors significantly improved first version of manuscript. It falls within the scope of the journal and can be accepted for publication.
Reviewer 3 Report
Dear authors,the manuscript is now much better written and suitable for publication. Please pay attention to minor grammatical errors and spaces between words, as well as citing references